# Interstitial Lung Disease in Immunocompromised Children

**DOI:** 10.3390/diagnostics13010064

**Published:** 2022-12-26

**Authors:** Xianfei Gao, Katarzyna Michel, Matthias Griese

**Affiliations:** Dr. von Haunersches Kinderspital, German Center for Lung Research (DZL), University of Munich, Lindwurmstr. 4a, D-80337 Munich, Germany

**Keywords:** interstitial lung disease, ILD, diffuse parenchymal lung disease, primary immunodeficiency, PID, secondary immunodeficiency, SID, genetic defect

## Abstract

Background: The range of pulmonary complications beyond infections in pediatric immunocompromised patients is broad but not well characterized. Our goal was to assess the spectrum of disorders with a focus on interstitial lung diseases (ILD) in immunodeficient patients. Methods: We reviewed 217 immunocompromised children attending a specialized pneumology service during a period of 23 years. We assigned molecular diagnoses where possible and categorized the underlying immunological conditions into inborn errors of immunity or secondary immunodeficiencies according to the IUIS and the pulmonary conditions according to the chILD-EU classification system. Results: Among a wide array of conditions, opportunistic and chronic infections were the most frequent. ILD had a 40% prevalence. Of these children, 89% had a CT available, and 66% had a lung biopsy, which supported the diagnosis of ILD in 95% of cases. Histology was often lymphocyte predominant with the histo-pattern of granulomatous and lymphocytic interstitial lung disease (GLILD), follicular bronchiolitis or lymphocytic interstitial pneumonitis. Of interest, DIP, PAP and NSIP were also diagnosed. ILD was detected in several immunological disorders not yet associated with ILD. Conclusions: Specialized pneumological expertise is necessary to manage the full spectrum of respiratory complications in pediatric immunocompromised patients.

## 1. Introduction

The lung is a complex parenchymal tissue ensuring proper gas exchange. While continuously perfused with blood through the capillary network, the large internal surface of the organ is exposed to air-born micro-organisms and many other environmental factors. A robust immunological balance is necessary to keep this delicate system fully functioning [1,2]. Host defense and multiple immunological, inflammatory and structural reactions involve, on the one hand, the airways contacting the outside world and, on the other hand, the interstitial organ compartment. These defense processes can resolve or lead to chronic immune cell-shaped specific tissue reactions, including fibrotic tissue repair processes or organ destruction with respiratory failure [3]. Due to this fragile balance of immune tolerance and response, it is obvious that the lungs are an important target organ in immunocompromised patients; pulmonary complications have been shown to represent the main clinical manifestations of immunodeficiencies and are an important cause of death [4,5,6]. Childhood immunodeficiencies are a broad group of rare diseases either caused by inborn errors of immunity, classified as primary immunodeficiency, or by hemato-oncologic diseases or immunosuppressive treatments leading to secondary immunodeficiency. The recent classification of the primary immunodeficiencies differentiates more than 400 different molecularly defined entities [7]. Such fine granular classification has not yet been used to address the frequency and type of different pulmonary complications in children.

In the past, the focus was predominantly on infectious pulmonary complications. However, lung disease may clinically not only manifest as airway disease, including bronchitis, bronchiectasis, obliterating bronchiolitis (BO) or asthma, but also as diffuse parenchymal or interstitial lung disease (ILD), including pulmonary hypertension (PHT) or lymphoproliferative disease (PTLD) [8]. Infrequently, pleural disease or pneumothorax is observed. Depending on the extent, all conditions may lead to respiratory failure with diffuse alveolar damage or acute respiratory distress syndrome (ARDS).

As both immunodeficiency and its pulmonary complications are rare, an overview from a specialized pediatric pneumology unit may be helpful to highlight some useful perspectives for the immunologist [9,10,11]. Our goal was to provide details on the clinical characteristics, including the age of onset, results of broncho-alveolar lavage, lung biopsy, chest computer tomography (CT), as well as the outcome of pediatric immunocompromised patients and pulmonary disease. Specifically, we focused on ILD manifestations within the different entities of immunodeficiency. Our findings indicate a high rate of various non-infectious complications and provide insight into the management of these patients in clinical practice.

## 2. Materials and Methods

We included all immunocompromised children assessed for significant lung disease between 1997 and 2020 in the Department of Pediatric Pneumology at the Dr. von Hauner Children’s Hospital of the University of Munich. Clinical information was collected retrospectively from the pneumological clinics’ charts, and patient files were updated for follow-up information. 

Data on gender, age at investigation, consanguinity, family history, gestational age, O_2_ supplement or mechanical ventilation during the neonatal period, as well as information on genetic and immunologic diagnostics and lung disease outcome were collected. Imaging studies were evaluated by pediatric radiologists with long-standing expertise in chest imaging, especially in pediatric interstitial lung diseases. Flexible bronchoscopy, including bronchoalveolar lavage (BAL), was performed if clinically indicated using 1 mL warmed normal saline per kilogram body weight 3 to 4 times [12]. BAL was performed in the most affected lobe or middle lobe if diffuse and examined for cell differentiation and microbiologically. In cases where a lung biopsy was obtained, the tissue was investigated by light microscopy, routine stain (hematoxylin and eosin stain (HE), Elastica van Gieson, PAS, iron) and bombesin, where indicated [13]. 

The immunodeficiencies were categorized using the system published by the International Union of Immunological Societies (IUIS) for inborn errors of immunity [7]. The primary immunodeficiencies included combined deficiencies, combined immunodeficiencies with syndromic features, antibody deficiencies, immune dysregulation, congenital defects of phagocyte number or function, defects of intrinsic and innate immunity, autoinflammatory syndromes and bone marrow failure. The secondary immunodeficiencies were due to malignancies or immunosuppressive treatment, including leukemias, lymphoma, other cancers, and transplantations. 

The pulmonary conditions were categorized by the updated etiologic classification system of the chILD-EU register [14]. Currently, the histopathological description of lung biopsies helps to categorize and distinguish specific parenchymal reaction patterns dominated by certain cell types or tissue components [15]. These most frequently include non-specific interstitial pneumonitis (NSIP), lymphoid interstitial pneumonitis (LIP), follicular bronchiolitis, granulomatous and lymphocytic interstitial lung disease (GLILD), desquamative interstitial pneumonitis (DIP) and alveolar proteinosis (PAP). NSIP histopathology consists of varying degrees of chronic inflammation and interstitial fibrosis, expanding the alveolar walls temporally and spatially uniformly and preserving lung architecture. The inflammation consists of lymphoid cells, mainly CD3-positive T lymphocytes, and small aggregates of CD20-positive B lymphocytes. The degree of infiltration is less than that in LIP, although potential overlap is recognized. Follicular bronchiolitis is characterized by lymphoid follicles around the bronchioles. DIP is characterized pathologically by a uniform involvement of lung parenchyma with an intra-alveolar accumulation of alveolar macrophages. Mild chronic lymphocytic inflammation and mild-moderate interstitial fibrosis may be present. PAP is a sometimes patchy intra-alveolar accumulation of amorphous, PAS-positive granular eosinophilic material that is lipid-rich (surfactant) and can contain cholesterol clefts and foamy macrophages [15].

## 3. Results

### 3.1. Characteristics of the Immunodeficiency Population and Spectrum of Associated Lung Diseases

The local pulmonary database retrieved 228 children, adolescents and young adults allocated to the disease category immunocompromised (Appendix A); 217 cases had sufficient information for review (Figure 1). Overall, more boys than girls were affected (60% vs. 40%), the majority (90%) of children were born as mature newborns, and less than 10% had respiratory problems at birth. Disease onset was at a median age of 2 years (Table 1).

A broad spectrum of lung diseases was identified (Table 2). Opportunistic and chronic infections were most frequent, occurring across all groups of immunodeficiencies at a rate of 65%. In the 129 BAL samples available from this group, viral, fungal and bacterial infections occurred. The most common opportunistic infections were caused by *Pneumocystis jirovecii* (12%). Cytomegalovirus was the second most common pathogen (5%). Bacteria, including *Viridans streptococci*, *Streptococcus pneumoniae* and *Haemophilus influenzae*, were also common causes of infection in this group (4%). Interstitial lung diseases were the second most common pulmonary complication, occurring at a rate of 40%. Respiratory failure was identified in more than 25% of the patients. Other less frequent conditions included ARDS, diffuse alveolar damage, pulmonary hypertension, bronchiolitis obliterans, bronchiectasis, PTLD, pneumothorax, asthma and pleural disease (Appendix A). 

Comparing primary and secondary immunodeficiencies, the frequency of bronchiolitis obliterans was higher in the latter, whereas opportunistic and recurrent infections were more frequently observed in the group of primary immunodeficiencies. Interestingly, ILD frequency was the same in both groups. Next, we focused on the group of immunocompromised children with ILD.

### 3.2. Comparison of Immunodeficient Children with and without ILD

The patients were divided into two groups: (1) those with immunodeficiency and ILD and (2) those with immunodeficiency without ILD. More than 40% of the immunodeficient children were diagnosed with ILD. No significant differences in the clinical characteristics were evident (Table 1), including the cellular composition of broncho-alveolar lavage (Table 3). The numerically higher percentage of eosinophils in the BAL fluid of patients with immunodeficiency and ILD might point towards immune dysregulation in those patients; however, the difference was not statistically significant. Overall, patients had elevated percentages of neutrophils (normal < 3%) and eosinophils (normal < 0.5%) in their lavages, independent of the presence of ILD. This differentiation was based on cytology results, as the immunophenotyping of BAL cells was not regularly conducted.

### 3.3. Features of ILDs in Immunodeficient Children

Within the group of patients who had developed an ILD, those with primary immunodeficiency more frequently had a family history of ILD and consanguinity (Table 4), pointing towards a potential genetic predisposition and risk factors for ILD. Gender distribution, age at disease onset, neonatal history and outcome of lung disease were not different when comparing primary and secondary immunodeficiency. 

In 90% of the children with ILD, a CT scan was performed, and in 80% of the studies, the features were consistent with an ILD (Table 5). Two-thirds of all children with ILD had a lung biopsy, which supported the diagnosis of ILD in 95% of cases. There were three ILD cases not supported by lung biopsy. Histopathological diagnosis in these patients included a normal transplanted lung, chronic bronchitis and a DAD with bronchiolitis obliterans. If genetic testing was performed, a monogenic condition known to be associated with ILD was identified in 76% of the patients. In more than two-thirds of the cases, the diagnosis of ILD was supported by two or three different diagnostic tests (Table 5).

The spectrum of histopathological ILD patterns in the lung biopsies of the immunodeficient patients was broad. Typical lymphocyte-dominated conditions were most prevalent and included GLILD, follicular bronchiolitis, LIP and NSIP, and constituted a total of 41% of all biopsies (Table 6). Other histological patterns included cholesterol pneumonia, DIP, PAP, lung fibrosis and pulmonary hemosiderosis, among others. Lung fibrosis was indicated in 13 patients, 3 of whom suffered from primary and 10 from secondary immunodeficiency (data not shown).

### 3.4. ILD in Genetically Defined Primary Immunodeficiency: Experience from a Single Pediatric Pneumology Center and Review of Literature

The frequency of ILD observed in patients with immunodeficiency and genetically identified causes observed in our cohort is depicted in Table 7. For comparison, we performed a literature review of genetically determined immunodeficiency conditions present in our cohort and extracted the associated pulmonary conditions (Table 7). Whereas opportunistic infections were the most frequently reported, ILD was prevalent in multiple but not all disorders. In 18 out of 25 conditions, we did not observe ILD involvement of the lungs in agreement with the literature, whereas, in 7 conditions, we observed an ILD. These diseases were caused by genetic variants in CD40, 10p13-p14DS, HELLS, TNFRSF13B, CYBA and NCF2. The patients in this group presented with an ILD-typical phenotype; however, susceptibility to opportunistic pathogens, including Cytomegalovirus, *Pneumocystis jirovecii* and *Aspergillus*, was coincidental, suggesting a possible role of microorganisms in the resulting lung disease. Of note, all these conditions were mainly described in single case reports or small series, increasing the likelihood that ILD manifestations might have been missed previously.

## 4. Discussion

Our data on lung diseases in immunodeficiencies confirmed that opportunistic and recurrent infectious diseases are still among the most prevalent pulmonary complications in an immunocompromised host; however, the data clearly demonstrate that formerly less frequently diagnosed conditions need to be considered carefully in clinical practice. This is particularly true for ILDs during childhood, which were identified in more than 40% of all patients. Of great interest and importance is an accurate etiological differentiation of the ILDs, as they represent an extremely broad spectrum of various disorders. Of note, many other but less frequent pneumological disorders, including bronchiolitis obliterans and pulmonary hypertension, must also be differentiated. 

There are several lessons to be learned from our study. (1) Respiratory complications in primary and secondary immunodeficiencies are important problems and need to be carefully addressed by clinicians; (2) the spectrum of pulmonary differential diagnosis beyond infectious complications is broad, including various forms of ILDs; (3) GLILD is a useful umbrella term alerting for ILD, but in immunodeficiencies, there are also other ILDs than GLILD; (4) traditional histopathological analysis can give important clues not only for differential treatments but also supporting advanced diagnostic multi-omics in the near future; (5) the limitation of cross-sectional analysis needs to be overcome by longitudinal studies, e.g., in registries to assess the course and stages of molecular entities with the help of CT imaging, lung function testing and deep clinical follow-up; and (6) importantly, close collaboration between immunologists and pulmonologists and other involved subspecialties will likely make an important difference.

Overall, 18% of the patients included died, and 15% became worse during the observation time. Even treatment patients with immunodeficiency still suffered from high rates of pulmonary infections (primary immunodeficiency 73%, secondary immunodeficiency 56%) or non-infectious chronic lung disease. While respiratory diseases started at a median age of about 2 years (range 0 to 20), neonatal respiratory disease was not a risk factor for later lung affection. Beyond suppurative infectious lung disease, various kinds of obstructive lung diseases, including bronchiolitis obliterans, spontaneous and recurrent pneumothorax, acute respiratory distress syndrome (ARDS), acute and chronic respiratory insufficiency, partial and global respiratory failure, and diffuse alveolar damage (DAD), were noted (see Table 2, Appendix A). Less frequent conditions to differentiate diagnostically in the wide spectrum of pulmonary affections were PTLD, subpleural and pleural fibrosis, pleurisy, pleural effusion, pleural empyema, pulmonary hypertension, portopulmonary hypertension, stenosis of the pulmonary artery, and pronounced obliterative vasculopathy.

For the pneumologist, ILD may be the presenting condition, and the underlying immunodeficiency is not yet diagnosed [92]. Several of the ILDs we identified in our sample are not typically expected in immunodeficiency, i.e., conditions not linked to GLILD. Such conditions included restrictive lung diseases such as cholesterol pneumonia, DIP, pulmonary hemosiderosis, pulmonary hemorrhage, bronchopulmonary dysplasia, PAP or NSIP. None of these histological patterns corresponded to a single disease entity. As an example, the NSIP pattern was found in connective tissue diseases, drug-induced ILD, hypersensitivity pneumonitis, HIV infection, chronic infection, chronic aspiration, previous acute lung injury and idiopathic NSIP [15].

All of these conditions are rare, and making such a diagnosis or not may contribute to the wide variation of ILD frequencies reported in several case series of immunocompromised children. While a recent pediatric study reported a very low rate of ILD, e.g., 1% (11/796 cases [93]), all other reports indicate higher rates (64% (39/61 pediatric and adult cases [94]), 15% [95], 34% [96], 26% (18/69 cases [97]), 11% (3/28 cases), 7% (46/637 cases [98]), 15% (8/54 cases [99]), 13% (78/623 cases [100]), 30% (24/80 cases [101]), 11% (40/370 cases [102]), 8% (6/73 cases [103]), 7% (114/1647 cases [104]), 10% (22/219 cases [105]), 40% (29/73 [106], 18% (9/50 cases [107]), 9% (138/1518 cases [108]), 22% (16/73 cases [109]), 20% (29/148 cases [110]), and 12% (4/33 cases [111]). It is clear that such differences result from selection bias due to differences in criteria for diagnosis, different age groups investigated, variable underlying diseases or selection bias from the researcher’s perspective and interest, i.e., observing primarily from an immunological or pneumological viewpoint, and also knowledge about the conditions and the existence of such complications. More exact estimates could be collected in population-based prospective studies using appropriate inclusion and exclusion criteria and case definitions.

In our cohort, lung biopsies were conducted at a relatively high frequency in 66% of the ILD patients. This was most likely due to a highly selected cohort of subjects with significant pulmonary problems, presenting after various diagnostic efforts and empirical therapeutic trials had been made. The biopsies led to an ILD diagnosis in 95% of the cases. A precise diagnosis may also be important for novel treatments, e.g., the presence of fibrosis in a biopsy may support treatment with anti-fibrotic drugs such as nintedanib or pirfenidone. 

Hurst et al., 2017 generated a consensus statement for CVID, introducing GLILD defined as a “distinct clinico-radio-pathological ILD occurring in patients with CVID, associated with a lymphocytic infiltrate and/or granuloma in the lung, and in whom other conditions have been considered and where possible excluded” [112]. As the authors pointed out later, there is still complex terminology for ILD in CVID and no consensus [113]. We believe that GLILD may be a useful umbrella term alerting for ILD in immunodeficiencies. Using the category of GLILD as a practical approach for currently available treatments also appears appropriate, as the ILD associated with immunodeficiencies often represents some form of benign lymphoproliferative pathology, and the ILD may simply be a manifestation of some immune dysregulation [112,114]. However, the traditional histopathological analysis as conducted here can give important additional diagnostic clues and, in the near future, may also support advanced diagnostic tissue-based multi-omics [115]. 

Chest CTs were performed in about 89% of the subjects, and 80% of these were consistent with an ILD. CT is a sensitive technique to detect ILD. This was further supported by a high rate of concordance with radiological findings and the results of lung biopsies. Histopathological examination confirmed a suspected ILD in 95% of cases. However, CTs cannot differentiate the type of ILD; thus, lung biopsies do not always appear to be redundant. On CT imaging, interstitial thickening, pulmonary fibrosis, pleuropulmonary elastosis or pleuroparenchymal fibroelastosis were the most common findings. 

An important strength of this study was the use of the advanced contemporary classification system for inborn errors of immunity, which focuses on distinct genetic disease categories. In our study, 49% of the patients with primary immunodeficiency had an underlying monogenic defect supporting their diagnosis. For seven conditions, we provided new evidence for ILD pulmonary manifestations. Another strength includes the collection of rare and clinically significant conditions, i.e., about 10 new cases annually over a period of more than 2 decades. However, this study was a cross-sectional analysis, and precise follow-up was lacking. Other limitations include its retrospective, single-center design and a selection of more severely affected patients submitted to our pediatric pneumology department. Longitudinal studies, e.g., in registries following the course of well-defined molecular entities, may use pre-structured CT imaging, lung function testing and deep clinical follow-up to overcome such shortcomings. Lastly, close collaboration between all involved subspecialties will likely make an important difference in unraveling the details of lung targeting in immunodeficiencies.

## Figures and Tables

**Figure 1 diagnostics-13-00064-f001:**
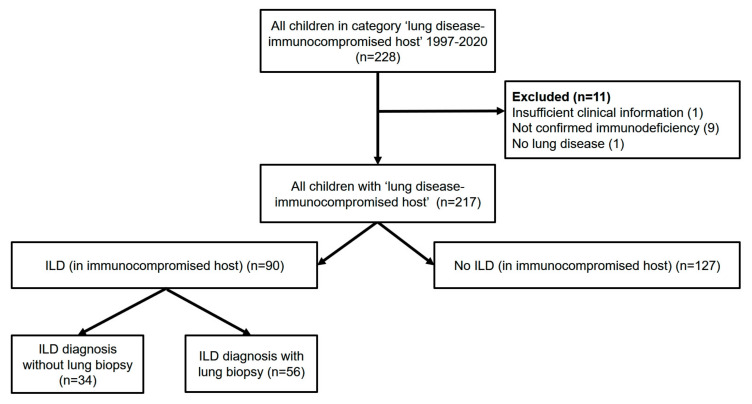
Overview of patients included and excluded.

**Table 1 diagnostics-13-00064-t001:** Clinical characteristics of patients with immunodeficiency and with ILD or without ILD.

	All Immunodeficiency Patients	Immunodeficiency with ILD	Immunodeficiency without ILD	Comparison between with/without ILD P
Total number	217	90 (41%)	127 (59%)	
Sex (male/female)	129 (59%)/88 (41%)	53 (59%)/37 (41%)	76 (60%)/51 (40%)	0.888 *
Age at onset of lung disease in years (range)	2.0 (0.0–20.1)	2.9 (0.0–15.2)	1.5 (0.0–20.1)	0.116 **
Follow-up duration in years (range)	4.9 (0.0–30.2)	4.9 (0.1–19.4)	4.9 (0.0–30.2)	0.893 **
ILD family history (yes/no)	10 (9%)/97 (91%)	8 (12%)/59 (88%)	2 (5%)/38 (95%)	0.233 *
Consanguinity (yes/no)	27 (25%)/79 (75%)	20 (30%)/46 (70%)	7 (18%)/33 (82%)	0.143 *
Gestational age (range)	40 (29–42)	40 (31–41)	40 (29–42)	0.696 **
O_2_ supplement in neonatal period (yes/no)	11/132	6/58	5/74	0.497 *
Mechanical ventilation in neonatal period (yes/no)	8/135	4/60	4/75	0.759 *
Outcome of lung disease at the end of follow-up				
Sick-better	96	39	57	0.853 *
Sick-same	46	23	23	0.177 *
Sick-worse	32	7	25	0.015 *
Dead	39	19	20	0.299 *

* Chi-square test, ** Mann–Whitney test.

**Table 2 diagnostics-13-00064-t002:** Immunodeficiency types and associated lung diseases.

	Lung Disease; n (% of Immunodeficiency Group)											
Immunodeficiency Type (n)	Interstitial Lung Disease	Pulmonary Hypertension	Infections (Opportunistic/Recurrent)	Bronchiolitis Obliterans	Bronchiectasis	PTLD	Respiratory Failure	ARDS	Diffuse Alveolar Damage	Pneumothorax	Asthma	Pleural Disease
All immunodeficiencies (217)	90 (41)	11 (5)	142 (65)	32 (15)	23 (11)	3 (1)	58 (27)	15 (7)	3 (1)	9 (4)	21 (10)	12 (6)
Primary immunodeficiencies (120)	52 (44)	6 (5)	88 (73)	6 (5)	19 (16)	-	31 (26)	9 (8)	-	1 (1)	13 (11)	4 (3)
Combined immunodeficiencies (22)	9 (41)	-	18 (82)	-	2 (9)	-	8 (36)	3(14)	-	1 (5)	1 (5)	-
Well-defined syndromes (15)	4 (27)	2 (13)	11 (73)	2 (13)	4 (27)	-	1 (7)	2 (13)	-	-	1 (7)	-
Antibody deficiencies (21)	4 (19)	1 (5)	17 (81)	1 (5)	8 (38)	-	2 (10)	1 (5)	-	-	6 (29)	-
Immune dysregulation (5)	2 (40)	-	3 (60)	1 (20)	-	-	1 (20)	-	-	-	-	-
Defects of phagocytes (30)	18 (60)	1 (3)	20 (67)	2 (7)	3 (10)	-	10 (33)	-	-	-	3 (10)	2 (7)
Defects of innate immunity (7)	3 (43)	-	6 (86)	-	1 (14)	-	4 (57)	2 (29)	-	-	-	1 (14)
Autoinflammatory syndromes (17)	10 (59)	2 (12)	12 (71)	-	1 (6)	-	4 (24)	1 (6)	-	-	2 (12)	1 (6)
Bone marrow failure (3)	2 (67)	-	1 (33)	-	-	-	1 (33)	-	-	-	-	-
Secondary immunodeficiencies (97)	38 (39)	5 (5)	54 (56)	26 (27)	4 (4)	3 (3)	27 (28)	6 (6)	3 (3)	8 (8)	8 (8)	8 (8)
ALL (15)	4 (27)	-	11 (73)	1 (7)	1 (7)	10(7)	5 (33)	1 (7)	-	-	1 (7)	1 (7)
AML (10)	3 (30)	-	6 (60)	1 (10)	-	-	3 (30)	-	1 (10)	-	-	1 (10)
Cancer, other (10)	4 (40)	-	4 (40)	1 (10)	-	-	-	1 (10)	-	1 (10)	1 (10)	-
CLL (2)	1 (50)	-	1 (50)	-	-	-	-	-	-	-	1 (50)	1 (50)
CML (1)	-	-	-	-	-	-	1 (100)	-	-	-	-	1 (100)
HIV (2)	1 (50)	-	1 (50)	-	-	-	-	-	-	-	-	-
Hodgkin lymphoma (3)	-	-	2 (67)	-	-	-	-	-	-	-	1 (33)	-
JMML (3)	2 (66)	1 (33)	2 (67)	1 (33)	-	-	1 (33)	1 (33)	-	-	-	-
MDS (5)	4 (80)	1 (20)	2 (40)	1 (20)	-	-	2 (40)	-	-	1 (20)	2 (40)	1 (20)
Non-Hodgkin lymphoma (1)	-	-	1 (100)	-	-	-	-	-	-	-	-	-
Other therap. intervention (1)	-	-	1 (100)	-	-	-	-	-	-	-	-	-
Transplant-heart (3)	2 (67)	-	2 (67)	-	-	1 (33)	-	-	-	-	-	1 (33)
Transplant-heart and lung (6)	3 (50)	3 (50)	2 (33)	2 (33)	-	-	2 (33)	-	1 (17)	-	-	-
Transplant-kidney (1)	-	-	1 (100)	-	-	-	1 (100)	-	-	-	-	-
Transplant-lung (4)	1 (25)	-	3 (75)	1 (25)	1 (25)	1 (25)	2 (50)	-	-	1 (25)	-	-
Transplant-stem cell (30)	14 (47)	-	15 (50)	18 (60)	2 (7)	1(3)	10 (33)	3 (10)	1 (3)	5 (17)	2 (7)	2 (7)

**Table 3 diagnostics-13-00064-t003:** Features of ILDs in immunodeficient children.

	All Immunodeficiency Patients	Immunodeficiency with ILD	Immunodeficiency without ILD	Comparison between with/without ILD P *
Cell concentration (/µL)	405.3 ± 1209.4 (114)	343.5 ± 308.1 (28)	425.4 ± 1383.0 (86)	0.216
Macrophages (%)	60.1 ± 27.9 (138)	64.7 ± 23.6 (46)	57.8 ± 29.6 (92)	0.299
PMN (%)	19.4 ± 25.9 (138)	16.3 ± 19.2 (46)	21.0 ± 28.7 (92)	0.626
Lymphocytes (%)	16.2 ± 17.0 (138)	13.5 ± 14.0 (46)	17.5 ± 18.2 (92)	0.437
Eosinophils (%)	2.1 ± 6.6 (138)	4.0 ± 10.5 (46)	1.2 ± 2.9 (92)	0.101
Mast cells (%)	0.2 ± 1.0 (138)	0.2 ± 0.8 (46)	0.3 ± 1.2 (92)	0.995
Plasma cells (%)	0.04 ± 0.3 (138)	0.03 ± 0.1 (46)	0.05 ± 0.4 (92)	0.388
Cell viability (%)	79.0 ± 23.0 (105)	80.4 ± 25.9 (23)	78.5 ± 22.2 (82)	0.283
Cell recovery (/µL)	53.0 ± 24.2 (107)	50.7 ± 28.5 (30)	53.9 ± 22.5 (77)	0.539 **

Data are means ± SD (n); * Mann–Whitney test, except “Cell recovery”, which was assessed with ** *t*-test.

**Table 4 diagnostics-13-00064-t004:** History, neonatal period and long-term course of patients with ILD and underlying immunodeficiency.

	Sex (m/f)	ILD Family History (y/n)	Consanguinity (y/n)	Age at Onset of Lung Disease (Years)	Gestational Age (Week)	O_2_ Supplement in Neonatal Period (y/n)	Mechanical Ventilation in Neonatal Period (y/n)	Follow-Up Duration (Years)	Outcome of Lung Disease			
									Sick-Better	Sick-Same	Sick-Worse	Dead
All types of immunodeficiency	53/37	8/59	20/46	4.4 ± 4.4	31–41 (40)	6/58	4/60	6.0 ± 5.2	39	23	7	19
Primary immunodeficiency	28/24	8/38	18/28	3.5 ± 4.2	31–41 (40)	4/38	2/40	6.8 ± 5.8	22	15	4	10
Combined immunodeficiencies	8/1	0/6	3/4	1.4 ± 3.2	34–41 (40)	0/8	0/8	2.4 ± 1.7	6	2	0	1
Well-defined syndromes	3/1	0/3	1/2	6.1 ± 5.5	37–40 (40)	1/1	1/1	3.6 ± 4.0	1	1	1	1
Antibody deficiencies	2/2	0/4	0/4	5.2 ± 6.9	37–40 (38)	0/4	0/4	6.3 ± 4.2	3	1	0	0
Immune dysregulation	2/0	0/2	0/2	2.7 ± 2.8	39–40 (40)	1/1	0/2	4.4 ± 3.8	1	0	1	0
Defects of phagocytes	4/14	3/13	9/6	4.5 ± 3.9	31–41 (40)	0/12	0/12	8.9 ± 5.2	8	5	2	2
Defects of innate immunity	2/1	2/1	3/0	0.6 ± 0.2	40 (40)	1/2	1/2	5.2 ± 8.1	0	0	0	3
Autoinflammatory syndromes	5/5	3/7	2/8	2.5 ± 2.8	34–40 (40)	1/8	0/9	9.4 ± 7.3	3	5	0	2
Bone marrow failure	2/0	0/2	0/2	7.7 ± 9.5	40 (40)	0/2	0/2	3.6 ± 4.0	0	1	.0	1
Secondary immunodeficiency	25/13	0/21	2/18	5.5 ± 4.4	32–40 (40)	2/20	2/20	5.0 ± 4.3	17	8	3	9
Comparisons between primary and secondary immunodeficiency	0.255 *	0.042 *	0.018 *	0.612 **	0.901 **	1.0 *	0.603 *	0.445 **	0.217 *	0.412 *	1.000 *	0.596

Data are numbers or means ± SD. Comparisons were made between primary and secondary immunodeficiency by * chi-square tests, ** ANOVA.

**Table 5 diagnostics-13-00064-t005:** ILD diagnosis supported by the results of the diagnostic tests used in the cohort of 90 patients with immunodeficiency.

	Numbers; %
Chest CT completed (yes/nk; % yes of all patients)	80/10; 89
ILD consistent with CT diagnosis (yes/no; % yes of patients with this test)	64/16; 80
Lung biopsy completed (yes/nk; % yes of all patients)	59/31; 66
Lung biopsy diagnosis proving ILD (yes/no; % yes of patients with this test)	56/3; 95
Genetic testing completed (yes/nk; % yes of all patients)	44/46; 49
Gene identified known to be associated with ILD (yes/no; % yes of patients with this test)	34/10; 76
ILD supported by genetics and lung biopsy (yes/no; % yes of patients with these tests)	21/28; 75
ILD supported by genetics and CT (yes/no; % yes of patients with these tests)	26/38; 68
ILD supported by lung biopsy and CT (yes/no; % yes of patients with these tests)	43/56; 77
ILD supported by genetics, biopsy and CT (yes/no; % yes of patients with these tests)	18/26; 69
ILD diagnosis only according to clinical records	3/87; 3

nk = not known or not available.

**Table 6 diagnostics-13-00064-t006:** Histopathological ILD diagnosis observed in 56 patients with immunodeficiencies and a lung biopsy.

Immunodeficiency (n, Percentage of Histopathological ILD Diagnosis in Immunodeficiency Subcategories)	Gender (Male/Female)	Histopathological Diagnosis and Pattern (n)
Primary immunodeficiency (32, 27%)		
Combined deficiencies (5)	4/1	NSIP (1), GLILD (1), Interstitial pneumonia (1), Intra-alveolar haemorrhage (1), Follicular bronchiolitis (1)
Well-defined syndromes (2)	2/0	Interstitial pneumonia (1), CPI (1)
Antibody deficiencies (2)	2/0	GLILD (1), Interstitial pneumonia (1)
Immune dysregulation (2)	2/0	LIP (2)
Defects of phagocytes (11)	1/10	Cholesterol pneumonia (1), DIP (2), PAP (7), Interstitial pneumonia (1)
Autoinflammatory syndromes (8)	4/4	LIP (1), Follicular bronchiolitis (1), NSIP (2), Interstitial pneumonia (1), DIP (1), PAP (1), Lung hypoplasia (1)
Bone marrow failure (2)	2/0	NSIP (1), Lung fibrosis (1)
Secondary immunodeficiency (24, 25%)		
ALL (3)	2/1	GLILD (1), NSIP (1), Follicular bronchiolitis (1)
AML (1)	1/0	PAP (1)
Cancer (2)	2/0	BPD (1), DIP (1)
JMML (2)	2/0	Follicular bronchiolitis (1), Pulmonary hemosiderosis (1)
MDS (2)	2/0	Lung fibrosis (1), NSIP (1)
Transplanted (14)	9/5	LIP (1), DIP + NSIP (1), DAD (1), Lung fibrosis (4), Cholesterol pneumonia (2), NSIP (5)

**Table 7 diagnostics-13-00064-t007:** Comparison between our cohort and literature regarding the percentage of the presence of ILD in genetically defined primary immunodeficiency.

Immunodeficiency Subcategories (Number of Patients with ILD/Number of Patients with Genetically Defined Immunodeficiency)	Disease Genetically Defined in Our Cohort (No.)	No. of Cases with ILD in Our Cohort (ILD Percentage)	Pulmonary Diseases Other than ILD (n)	Prevalence of ILD (%) in Primary Immunodeficiency Genetic Defect, as Reported in the Literature (May 1999 to May 2022)	Gene Identified, Known to Be Associated with a Condition Presenting with an ILD
Combined deficiencies (4/8)	ADA (2)	1 (50%)	ARDS, Respiratory failure, Opportunistic/recurrent infections (1)	44% [16] **	Y
	CD40 (2)	2 (100%)		0% [17] **, [18] *	N
	CD40LG (1)	0 (0%)	Opportunistic/recurrent infections (1)	20% [19] ***, 0% [20] ***	N
	IL2RG (1)	1 (100%)		7% ILD [21] ***	Y
	DOCK8 (1)	0 (0%)	Bronchiolitis obliterans, Bronchiectasis (1)	0% [22] ***, [23] ***	N
	RFXAP (1)	0 (0%)	Opportunistic/recurrent infections (1)	50% [24] *	Y
Well-defined syndromes (4/14)	ATM (3)	1 (33%)	Bronchiectasis, Pulmonary hypertension (1), Bronchiectasis, Opportunistic/recurrent infections (1)	50% [25] *, 26% [26] ***, 14% [27] ***	Y
	10p13-p14DS (1)	1 (100%)		0% [28] ***, [29] *	N
	MCM4 (1)	1 (100%)		50% [30] **, 0% [31] **	Y
	DNMT3B (1)	0 (0%)	Opportunistic/recurrent infections (1)	0% [32] ***	N
	IKBA (1)	0 (0%)	Opportunistic/recurrent infections (1)	0% [33] *, [34] **	N
	NBS1 (1)	0 (0%)	Bronchiectasis, Opportunistic/recurrent infections (1)	0% [35] *. [36] *, [37] ***	N
	TTC7A (1)	0 (0%)	Opportunistic/recurrent infections (1)	0% [38] *, [39] *, [40] ***	N
	DiGeorge (4)	0 (0%)	Opportunistic/recurrent infections (2), Asthma, Pulmonary hypertension (1), Opportunistic/recurrent infections, ADRS (1)	0% [41] ***, 100% [42] *	Y
	HELLS (1)	1 (100%)		0% [43] **, [44] *	N
Antibody deficiencies (1/4)	TNFRSF13B (1)	1 (100%)		0% [45] **, [46] *	N
	BTK (1)	0 (0%)	Opportunistic/recurrent infections, Respiratory failure (1)	0% [47] *, [48] ***	N
	NFKB1 (1)	0 (0%)	Opportunistic/recurrent infections, ARDS (1)	12% [49] ***, 0% [50] **	Y
	PIK3CD (1)	0 (0%)	Bronchiectasis, Opportunistic/recurrent infections (1)	0% [51] **, [52] **	N
Immune dysregulation (2/5)	FOXP3 (2)	1 (50%)	Bronchiolitis obliterans, Respiratory failure (1)	23% [53] ***,	Y
	STAT3 GOF (1)	1 (100%)		36% [54] ***, 100% [55] *	Y
	IL10 (1)	0 (0%)	Opportunistic/recurrent infections (1)	0% [56] **, [57] ***	N
	UNC13D (1)	0 (0%)	Opportunistic/recurrent infections (1)	0% [58] **, [59] *	N
Defects of phagocytes (18/21)	CSF2RA (15)	15 (100%)		100% [60] **, [61] ***	Y
	CYBA (3)	2 (67%)	Bronchiolitis obliterans, Opportunistic/recurrent infections, Asthma (1)	0% [62] **, [63] *, [64] ***, [65] ***	N
	CYBB (1)	0 (0%)	Opportunistic/recurrent infections (1)	0% [62] ***, [63] ***, [64] ***, [65] ***	N
	NCF4 (1)	0 (0%)	Bronchiectasis (1)	0% [65]*	N
	NCF2 (1)	1 (100%)		0% [62] **, [63] *, [64] ***, [65] ***	N
Defects of innate immunity (3/7)	MDA5 def (LOF). IFIH1 (1)	0 (0%)	Respiratory failure (1)	0% [66] *, 100% [67] *	Y
	STAT1 (AD LOF) (1)	0 (0%)	Bronchiectasis, Opportunistic/recurrent infections (1)	5% [68] ***	Y
	TCIRG1 (1)	0 (0%)	Opportunistic/recurrent infections (1)	0% [69]***	N
	ZNFX-1 (4)	3 (75%)	Opportunistic/recurrent infections (1)	13% [70] **, 50% [71] *	Y
Autoinflammatory syndromes (10/16)	COPA (7)	7 (100%)		100% [72] ***, [73] ***	Y
	OAS1 (1)	1 (100%)		100% [74] **, [75] *	Y
	PLCG2 (1)	1 (100%)		0% [76] **, [77] *, [78] ***	N
	STING (1)	1 (100%)		100% [79] ***, 85% [80] **	Y
	AGS7.IFIH1 (1)	0 (0%)	Opportunistic/recurrent infections, Respiratory failure, ARDS (1)	0% [81] *, [82] *, [83] **	N
	MEFV (2)	0 (0%)	Opportunistic/recurrent infections (2)	0% [84] ***, [85] ***, [86] ***	N
	TMEM173 (1)	0 (0%)	Opportunistic/recurrent infections (1)	100% [79] ***, 85% [80] **	Y
	TNFRSF1A (2)	0 (0%)	Asthma (1), Opportunistic/recurrent infections (1)	0% [87] ***	N
Bone marrow failure (2/3)	SAMD9 (1)	0 (0%)	Opportunistic/recurrent infections, Respiratory failure (1)	0% [88] **, [89] **	N
	TERC (1)	1 (100%)		50% [90] *	Y
	TERT (1)	1 (100%)		16% [90] ***, 56% [91] **	Y

* Case reports on 1 to 5 patients, ** Cohorts with 6 to 20 patients, *** Cohorts with more than 20 patients.

## Data Availability

The datasets generated during and/or analyzed during the current study are available from the corresponding author upon reasonable request.

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
