# Peer review of "Interstitial Lung Disease in Immunocompromised Children"

_diagnostics, 2022, doi:10.3390/diagnostics13010064_

Round 1
Reviewer 1 Report
I congratulate the authors, the work is well structured and scientifically valid. I find it very interesting given the serious consequences that interstitial lung diseases can cause. I have only a few minor considerations.
Minor questions:
1) In the Results section the authors state that chronic and opportunistic infections are the most frequent pathologies found among all groups of immunodeficiencies (lines 120-122). however I would suggest the authors add at least two or three lines of description listing which mycoorganisms cause the most frequent occurrence of infection in the study subjects (example: haemophilus influenzae, Staphylococcus aureus, respiratory syncytial virus, etc.).
2) In lines 142-150 speaking of the cellular composition of the broncholavage liquid the authors highlight that there is no significant difference in the cellular composition of the BAL between ILD and non-ILD subjects with immunodeficiency. However, given the importance of immune cell infiltration and the inflammatory response in the mechanism of lung tissue damage, in my opinion the authors should spend a few more comments on this data. In addition, it would be important (if data exist) to further investigate the cellular composition of BAL. That is, I would ask the authors to add data on which lymphocytes (CD3, CD3+CD4+, CD3CD8+, NK cells, B lymphocytes, CD38+HLA-DR+ T lymphocytes) are present in the BAL (provided that the immunophenotype on the BAL has been performed). Because it would be a very interesting.
3) In the same way, if there is data on the immunophenotype made on peripheral blood, this data should be added, with a table similar to table Number. 3, but relating to data on the immunophenotype from peripheral blood. (if the exam was performed).
4) In lines 202-206 the authors describe seven subjects with monogenic immunodeficiency and ILD, listing the mutated genes. I would suggest the authors add whether these seven patients were prone to infections and if so from which microorganisms. Emphasizing the possible presence of microorganisms with pulmonary tropism.
Author Response
We thank this reviewer for the important comments, which we have addressed as described below with the appropriate changes in the manuscript
C1) In the Results section the authors state that chronic and opportunistic infections are the most frequent pathologies found among all groups of immunodeficiencies (lines 120-122). however I would suggest the authors add at least two or three lines of description listing which mycoorganisms cause the most frequent occurrence of infection in the study subjects (example: haemophilus influenzae, Staphylococcus aureus, respiratory syncytial virus, etc.).
R1 The microorganism recovered from BAL were available in 129 cases. The highest rank had Pneumocystis jirovecii pneumonia with a frequency of 12% (16/129), and the second was Cytomegalovirus (CMV) with a frequency of 5% (7/129), the third ranks were Viridans streptococci, Haemophilus influenza and Streptococcus pneumonia with a frequency of 4% (5/129).
The paragraph (starting at line 120) was revised accordingly.
C2) In lines 142-150 speaking of the cellular composition of the broncholavage liquid the authors highlight that there is no significant difference in the cellular composition of the BAL between ILD and non-ILD subjects with immunodeficiency. However, given the importance of immune cell infiltration and the inflammatory response in the mechanism of lung tissue damage, in my opinion the authors should spend a few more comments on this data. In addition, it would be important (if data exist) to further investigate the cellular composition of BAL. That is, I would ask the authors to add data on which lymphocytes (CD3, CD3+CD4+, CD3CD8+, NK cells, B lymphocytes, CD38+HLA-DR+ T lymphocytes) are present in the BAL (provided that the immunophenotype on the BAL has been performed). Because it would be a very interesting.
R2 Unfortunately, immunophenotypization was not consistently conducted on BAL fluid, hence no lymphocyte differentiation data are available. We recognize the limitation of the study, this aspect should be further investigated in the future. This limitation is indicated in the manuscript in line 156.
C3) In the same way, if there is data on the immunophenotype made on peripheral blood, this data should be added, with a table similar to table Number. 3, but relating to data on the immunophenotype from peripheral blood. (if the exam was performed).
R3 The exam of immunophenotype from peripheral blood was unfortunately not available in this project.
C4) In lines 202-206 the authors describe seven subjects with monogenic immunodeficiency and ILD, listing the mutated genes. I would suggest the authors add whether these seven patients were prone to infections and if so from which microorganisms. Emphasizing the possible presence of microorganisms with pulmonary tropism.
R4 PCP, CMV, Haemophilus infuenzae, and Aspergillus were the microorganisms which induced the infection in those patients (table below).
|
Gene |
Microorganism present in the affected patient |
|
TNFRSF13B |
Unknown |
|
NCF2 |
CMV, Haemophilus influenzae |
|
HELLS |
PJP |
|
CYBA |
Aspergillus |
|
CD 40 |
CMV |
|
CD 40 |
PJP |
|
10p13-p14DS |
unknown |
This information is now included in the paper

Reviewer 2 Report
In this article, Gao et al reported on 217 immunocompromised children in detail. Few studies have described in such detail, and I believe that this is a very valuable report. In particular, the relationship between immunodeficiency and ILD is reported in detail, suggesting the involvement of immunological mechanisms in the development of ILD. However, this is a cross-sectional study, and it is unfortunate that the follow-up was not thorough. Another limitation is that it is a retrospective, single-center study.
In light of the above, I have several questions.
1) In Table 1, the outcome of lung disease at the end of follow-up is significantly higher in patients without ILD compared to those with ILD.
Our first impression is that the disease is more severe with ILD.
Why is this?
2) In Table 4, primary immunodeficiency is significantly higher for consanguinity and family history of ILD. In the case of consanguineous marriages, this may be due to recessive inheritance by the carrier, etc., but what do you think is the reason for the significant increase in family history of ILD?
Is this related to question 1) above?
3) In these 217 patients, do you know, for example, what kind of treatment was given to those with ILD and those without ILD, respectively?
Author Response
C1) In Table 1, the outcome of lung disease at the end of follow-up is significantly higher in patients without ILD compared to those with ILD. Our first impression is that the disease is more severe with ILD. Why is this?
R1 The ILD adds an additional, sometimes difficult to trat condition to a patients with an immuno deficiency. Often ILD may be progressive on its own (Doubkova et al 2015, Schussler et al 2016).
C2) In Table 4, primary immunodeficiency is significantly higher for consanguinity and family history of ILD. In the case of consanguineous marriages, this may be due to recessive inheritance by the carrier, etc., but what do you think is the reason for the significant increase in family history of ILD? Is this related to question 1) above?
R2 This question is difficult to answer. We believe that consanguinity is one marker for genetically caused conditions and family history another one, may be weaker, as there is some dilution by sporadic cases. As the ILD is considered as a part of process of PID and PIDs may be caused by mutations and can have a significant increase in family history, the ILD is also related to genetic mutations and increase in family history.
C3) In these 217 patients, do you know, for example, what kind of treatment was given to those with ILD and those without ILD, respectively?
R3 We know the drug categories used for treatment of these 217 patients. Patients with ILD and without ILD received similarly antibiotics/antimicrobials, steroids, immunoglobulins, O2 supply and immunosuppressive drugs.
